# Developing political-ecological theory: The need for many-task computing

**Timothy Haas** [ORCID] *

Lubar School of Business, University of Wisconsin-Milwaukee, Milwaukee, WI, United States of America

* haas@uwm.edu

## Abstract

Models of political-ecological systems can inform policies for managing ecosystems that contain endangered species. To increase the credibility of these models, massive computation is needed to statistically estimate the model's parameters, compute confidence intervals for these parameters, determine the model's prediction error rate, and assess its sensitivity to parameter misspecification. To meet this statistical and computational challenge, this article delivers statistical algorithms and a method for constructing ecosystem management plans that are coded as distributed computing applications. These applications can run on cluster computers, the cloud, or a collection of in-house workstations. This downloadable code is used to address the challenge of conserving the East African cheetah (*Acinonyx jubatus*). This demonstration means that the new standard of credibility that any political-ecological model needs to meet is the one given herein.

**Data Availability Statement:** All relevant data are within the manuscript and its Supporting information files.

**Funding:** The author(s) received no specific funding for this work.

## Introduction

There is a need to acknowledge the complexity of political-ecological systems and the significant challenges to building theories of them [1–3]. Such systems lie at the interface between social/political science and ecology. The complexity of each of these fields coupled with an additional layer of complexity introduced by the interactions between sociological/political systems and natural systems can result in highly complex system dynamics, i.e., ones that are stiff, nonlinear, and possess feedback loops. For example, Schoon and Van der Leeuw [4] note that systems composed of interacting sociological and ecological subsystems are quick to change and rarely stay in equilibrium for long. Further, many state variables are needed to describe both the decision making processes of the relevant social groups, and the functioning of the involved ecosystem. A *political-ecological system* is also referred to as a *socio-ecological* system or *social-ecological* system (e.g., see [5]). The former term is emphasized herein because those political actions and processes that drive social movements are often initiated by groups seeking to gain increased political power [6]. The decline in the planet's biodiversity [7], creates a need for credible political-ecological theory to guide the development of sustainable biodiversity conservation policies.

In addition to the challenge of building political-ecological theory, there is a deeper problem with using such models to guide ecosystem management policy: Unless such a model is

**Competing interests:** The authors have declared that no competing interests exist.

shown to be credible, any policy recommendations based on output from the model may receive only mixed acceptance by those affected. As argued in [8], there is a need for a common model credibility standard to be met before the output of a model of a political-ecological system is deemed to be policy-relevant. This is because there may be skepticism towards models that have not had their parameters statistically estimated nor their parameter sensitivities assessed [9, 10]. These skeptics may be unwilling to cooperate with efforts to implement ecosystem management policies that are based in-part on output from these unassessed models.

But what is a credible model? Patterson and Whelan [11] state that "Model credibility is about the willingness of people to make decisions based on the predictions from the model." In other words, a model is credible when a decision maker places enough trust in its predictions to use those predictions to select management actions. Call the model's behavior, functioning, relationships, and systems of equations, its collective *mechanism*. Patterson and Whelan [11] believe the decision maker's trust is won if (a) the model's mechanism is based on known principles that govern the phenomenon being modeled; (b) all aspects of the model's mechanism are testable, i.e., there are observable variables in the model on which data may be collected and used to conduct statistical hypothesis tests of the presence of these behaviors in the real world; and (c) the out-of-sample prediction error of the model's predictions is below the decision maker's threshold.

To make the assessment of a political-ecological model's credibility easier to perform, this article develops and demonstrates an integrated suite of statistical methods for assessing model credibility components (b) and (c), above. Some of the hypotheses of component (b) may concern the sensitivity of the model to perturbations to its parameters. The testing of such hypotheses is typically referred to as performing a *sensitivity analysis*.

For the remainder of this article, the term "model validation" will not be used because in this author's opinion, it is too ambiguous a term to support a consensus about whether a valid model can be established at all, let alone how it might be quantitatively assessed (see [12] and [13]).

An *agent-based model* consists of a collection of entities that make a sequence of decisions through time based on their goals and inputs from other agents. An ABM is often built to model a social system that is too complex to represent using mathematical models [14]. In ecology, the word "agent" is often replaced with the word "individual" to emphasize that the entities are individual flora or fauna whose behavior is more genetically defined rather than being based on a belief system such as utility maximization. As the authors of [15] state, *individual-based models* (IBMs) "explicitly represent discrete individuals within an (ecological) population and their individual life cycles." One approach to modeling a political-ecological system is with a combination of an ABM to capture the system's anthropogenic actions, and an IBM to capture the dynamics of the affected ecosystem. These two submodels interact with each other in order to capture the effects of actions taken by groups of humans that affect the ecosystem—and the feedback effects from the ecosystem back to those groups.

For example, Haas and Ferreira [16] build an economic-ecological model of the rhinoceros (*Ceratotherium simum*) horn trafficking system. This model contains submodels (agents) of rhino horn consumers, rhino poachers, and those antipoaching units attempting to stop the poachers. These latter two submodels interact with an IBM of the rhino population being illegally harvested. Haas and Ferreira [17] extend the poachers group submodel of this ABM-IBM model by adding a mechanism that explains how these individuals weigh the risk of being prosecuted for poaching against its profit potential. These authors then use this submodel to evaluate the practicality of policies aimed at providing employment opportunities for rhino poachers versus policies that intensify the enforcement of anti-poaching laws. This ABM-IBM model contains several hundred parameters.

## Simulating a political-ecological system

**Definition:** A *political-ecological system simulator* (hereafter *simulator*) is an executable computer program capable of approximating the outputs of a stochastic model of a political-ecological system.

Such a simulator is part of an *ecosystem management tool* (EMT) developed by Haas [8]. An EMT is used to find politically feasible and effective policies for managing at-risk ecosystems. In this simulator, influence diagrams (IDs) (see [18]) are used to implement submodels for group decision making, and ecosystem functioning. For instance, the political-ecological system models of Haas and Ferreira [16, 17, 19] are computationally implemented through their attendant simulators.

This article's central argument is that for simulators to effectively contribute to the development of political-ecological theory and ecosystem management policies, the following three activities need to be performed in sequence: (1) statistically fitting the simulator's parameters to data sets of *political-ecological actions* [20], (2) assessing the credibility of this fitted simulator, and (3) running computations on this (now) credible simulator to find politically feasible and sustainable ecosystem management policies.

## Addressing the computational challenge

Call one execution of a command to statistically estimate the parameters of a model, a *job* (see [21] and [22]). Generalizing this idea, let a *simulator job* refer to one execution of the computations needed to either (1) statistically estimate the parameters of a political-ecological system simulator; (2) compute parameter confidence intervals; (3) compute a measure of a simulator's prediction error rate; (4) perform a sensitivity analysis; or (5) find, using the simulator, an ecosystem management policy. These five simulator jobs are integrated in that the first two jobs share the same estimator, the fourth job needs the confidence intervals found in the second job, and the fifth job uses the fitted model that was found by the first job.

Simulator jobs can require large amounts of computer time. From now on, however, the use of policy-relevant statistical and optimization methods will be possible only if the attendant computational challenges are met. Hence, any discussion or evaluation of such methods is inseparable from a consideration of their computational implementations.

But the need for large amounts of computer time can become a challenge for those scientists, government agencies, and NGOs needing to run such computations. Hereafter, call these groups and individuals who are involved in biodiversity protection, *ecosystem managers*. The handicap these managers face is that funding to support the active management of ecosystems can be uneven. For example, circa 2017-2020, the United States Environmental Protection Agency (USEPA) is being down-sized by President Trump's administration [23]. But managing an ecosystem with the goal of conserving its biodiversity requires an on-going analysis of monitoring data as it arrives in order to guide the development of management actions that, when implemented, result in successful biodiversity outcomes. This means that ecosystem managers need to have alternative computing options should they be temporarily unable to afford supercomputer time from an external *high performance computing* (HPC) provider.

This article argues that a practical way to meet this computational challenge is to implement these jobs as *many-task computing* (MTC) applications. The authors of [24] describe such jobs as being made up of a collection of within-job computations, called *tasks* that are loosely coupled, communication-intensive, and heterogeneous. Several *application program interfaces* (APIs) that can be used to implement such jobs are described below, and one, JavaSpaces™ (see [25]) is demonstrated through a case study.

### Article contributions

This article makes three contributions to the development of political-ecological theory and the use of such theory in the formation of ecosystem management policies:

1. the first integrated suite of statistical measures for performing parameter estimation and credibility assessment of a political-ecological model and its attendant simulator,

2. a new method for constructing politically feasible and sustainable ecosystem management policies, and

3. downloadable software for implementing these methods as MTC applications via the JavaSpaces API.

## Related work

### Models, estimation, and sensitivity analysis

In a highly cited article, Macy and Willer [26] discuss how ABMs can advance sociological theory. Conte and Paolucci [27] note the potential that ABMs have for social science theory construction.

Methods exist for the statistical estimation of a socio-ecological model's parameters [17, 28]—and for the estimation of a deterministic ecological model [29–31]. *Minimum simulated distance estimators* (MSDEs) are one family of parameter estimators that can be used to estimate the parameters of a stochastic ecosystem model. And one way to define the needed distance function is with the Hellinger distance [32, 33]. For example, in [28], a Hellinger distance-based MSDE is used to estimate the parameters of a stochastic, dynamic model of a political-ecological system.

A model is sensitive to a set of parameters if small perturbations to their values significantly affect the model's outputs. For instance, the authors of [34] perform a *probabilistic sensitivity analysis* [35] of a salmon population dynamics model. And in [36], a probabilistic sensitivity analysis of an agricultural model is performed.

### Integrated statistical assessment of a socio-ecological model's credibility

A literature search uncovered no articles describing an integrated statistical assessment of a socio-ecological model's credibility. In [37], however, a specific suite of activities is given for statistically assessing an ecosystem model's credibility. These authors believe the evaluation of an ecosystem model should include (1) an interrogation of the model's logic to determine whether it is parsimonious and biologically realistic; (2) a statistical estimate of its parameters; (3) estimates of its prediction accuracy; (4) computation of statistical goodness-of-fit tests; and (5) a probabilistic sensitivity analysis. These authors, however, do not apply their recommendations to a case study.

Yarkoni and Westfall [38] call for a shift in focus from building models that pass in-sample goodness-of-fit (GOF) tests towards the building of models that have low prediction error rates (out-of-sample performance). This is particularly true for models that are used to guide decisions aimed at changing the future behavior of a system (out-of-sample). A political-ecological system is, in-part, a model of how humans behave and hence, the focus on prediction for psychological models as advocated by Yarkoni and Westfall applies to political-ecological models.

## Materials and methods

First, the procedure for using the EMT is given. This is followed by the statistical theory underpinning each simulator job. The section concludes with algorithms and runtime issues particular to the casting of simulator jobs as MTC applications.

## EMT procedure

The three activities of statistically fitting a simulator, assessing its credibility, and using it to find politically feasible and ecologically effective policies form part of a step-by-step procedure given in [8, pp. 77-78] for using the EMT. A new version of this procedure follows.

Step 1: Identify the spatial boundaries of the ecosystem to be managed. Typically, this ecosystem will host one or more endangered species.

Step 2: Identify those political groups that directly or indirectly affect this ecosystem. Construct submodels of these groups by casting them as IDs and expressing them in the **id** language. This language is part of the **id** software system (see [39]). Use theories of cognitive processing to assign *hypothesis values* to the parameters of these submodels. Load these values into *hypothesis parameter files*—one file for each group.

Step 3: Construct a population dynamics submodel of all species identified in Step 1. Cast this submodel as an ID and express it in the **id** language. Use ecological theory to identify hypothesis values for the parameters of this submodel. Load these values into a hypothesis parameter file.

Step 4: Using all of the above files, create a master file that defines the political-ecological system simulator.

Step 5: Acquire a data set of political-ecological actions made by some of the groups modeled in Step 2, and the ecosystem modeled in Step 3. The ecological component of this data set might consist of observations on the spatio-temporal abundance of several species.

Step 6: Use **id** to statistically fit some subset of the simulator's parameters to this data set using *consistency analysis* (see [28], and [8, pp. 46-52]).

Step 7: Use **id** to compute jackknife confidence intervals for the parameters estimated in Step 6.

Step 8: Conduct an analysis of the simulator's credibility (see [8, pp. 179-198]) by using **id** to perform the two separate jobs of (a) estimating the simulator's prediction error rate through computation of its one-step-ahead prediction error rates; and (b) performing a *deterministic sensitivity analysis* using thresholds defined by the parameter confidence intervals found in Step 7. If the simulator displays error rates that are no better than blind guessing (all options in each group submodel are equally likely), or it displays unacceptable sensitivity to some of its parameters, re-formulate one or more of the simulator's submodels and go back to Step 6. Continue in this manner until the simulator is credible.

Step 9: Use **id** to run a job with this (now) credible simulator to construct the *most practical ecosystem management plan* (MPEMP) (see [8, pp. 52-53]).

Step 10: Implement this MPEMP in the real world.

Step 11: As new data becomes available, repeat Steps 6 through 10.

## Statistical estimation of simulator parameters

The consistency analysis statistical estimator delivers parameter estimates that result in the simulator's probability distributions on its output variables being as similar as possible to

empirical distributions derived from data while at the same time being as close as possible to those derived from political-ecological theory. Consistency analysis is a parameter estimator that is related to MSDE.

**Hellinger distance.**   Following [28, Appendix], and [17, S3 Appendix], one way to define the distance between two multivariate probability distributions is as follows. Partition a vector of $p$ random variables, $\mathbf{U}$ into $\mathbf{U}^{(d)}$, and $\mathbf{U}^{(ac)}$—the vectors of discrete and absolutely continuous random variables, respectively. Say there are $d$ discrete members of $\mathbf{U}$, and $c$ continuous members. Hence, $p \equiv d + c$. Let the *probability density probability function* (PDPF) be

$$pf_{\mathbf{U}}(\mathbf{u}) \equiv \frac{\partial}{\partial \mathbf{U}^{(ac)}} P(\mathbf{U}^{(d)} = \mathbf{u}^{(d)},\ \mathbf{U}^{(ac)} \leq \mathbf{u}^{(ac)}). \tag{1}$$

Let $\mathbf{U}|\boldsymbol{\beta}$ notate the random vector whose PDPF is parameterized by the components of $\boldsymbol{\beta}$. For example, an ID might be composed of $U_1 \sim Bernoulli(\beta_1)$ and $U_2 \sim Normal(\beta_2 + u_1\beta_3, \beta_4)$. The graph of this ID appears in Fig 1, and its parameter vector, $\boldsymbol{\beta} = (\beta_1, \beta_2, \beta_3, \beta_4)$.

In terms of the PDPF, the Hellinger distance between two probability distributions is

$$\Delta(\beta_1, \beta_2) \equiv \frac{1}{\sqrt{2}} \left[ \int_{\mathbf{u}} \left( \sqrt{pf_{\mathbf{U}|\beta_1}(\mathbf{u}_i)} - \sqrt{pf_{\mathbf{U}|\beta_2}(\mathbf{u}_i)} \right)^2 \mathbf{du} \right]^{1/2} \tag{2}$$

and is bounded between 0 and 1 [40].

**Consistency analysis.**   Haas and Ferreira [17] give a description of consistency analysis before applying it to a model of the political-ecological system of rhino horn trafficking. An abbreviated version of this description appears here.

Let $m$ be the number of interacting IDs in a political-ecological simulator. Let $\mathbf{U}_i$ be the vector that contains all of the chance nodes that make up the $i^{th}$ ID (either one of the group submodels or the ecosystem submodel). Let $\mathbf{U}|\boldsymbol{\beta}_{(ij)}$ be the $i^{th}$ ID's probability distribution parameterized by the entries in $\boldsymbol{\beta}_{(ij)}$ under the $j^{th}$ set of conditioning (input) node values. Each parameter in the ID is assigned a point value a-priori that is derived from either expert opinion, subject matter theory, or the results of a previous consistency analysis. Collect all of these hypothesis values into the *hypothesis parameter vector*, $\boldsymbol{\beta}_H^{(ij)}$. This vector holds the ecosystem manager's prior beliefs about the true values of the model's parameters.

Let $l_i$ be the number of belief networks formed by conditioning the $i^{th}$ ID on all possible combinations of its input nodes. There are $m - 1$ group submodels, and one ecosystem submodel. Define

$$
\begin{aligned}
\mathcal{B}^{(Grp)} &\equiv (\beta^{(1,1)'}, \ldots, \beta^{(1,l_1)'}, \ldots, \beta^{(m-1,1)'}, \ldots, \beta^{(m-1,l_{m-1})'})', \\
\mathcal{B}^{(Eco)} &\equiv (\beta^{(m,1)'}, \ldots, \beta^{(m,l_m)'})',\ \text{and} \\
\mathcal{B} &\equiv (\mathcal{B}^{(Grp)'},\ \mathcal{B}^{(Eco)'})',
\end{aligned}
$$

i.e., those parameters that identify all of the group submodels, those that identify the ecosystem submodel, and the collection of all of the model's parameters, respectively.

As in [8, pp. 17-18], for group submodels, let an *in-combination* be a set of values on the input nodes {*time, input action, actor, subject*}. Let an *out-combination* be a set of values on the input nodes {*output action, target (of that action)*}. A group ID selects an out-combination by computing the expected value of its terminal node, `Overall Goal Attainment` under the received (given) in-combination—and each possible combination of values on the two input nodes of `Out-Action` and `Target`. The out-combination that maximizes this expected value is selected for output.

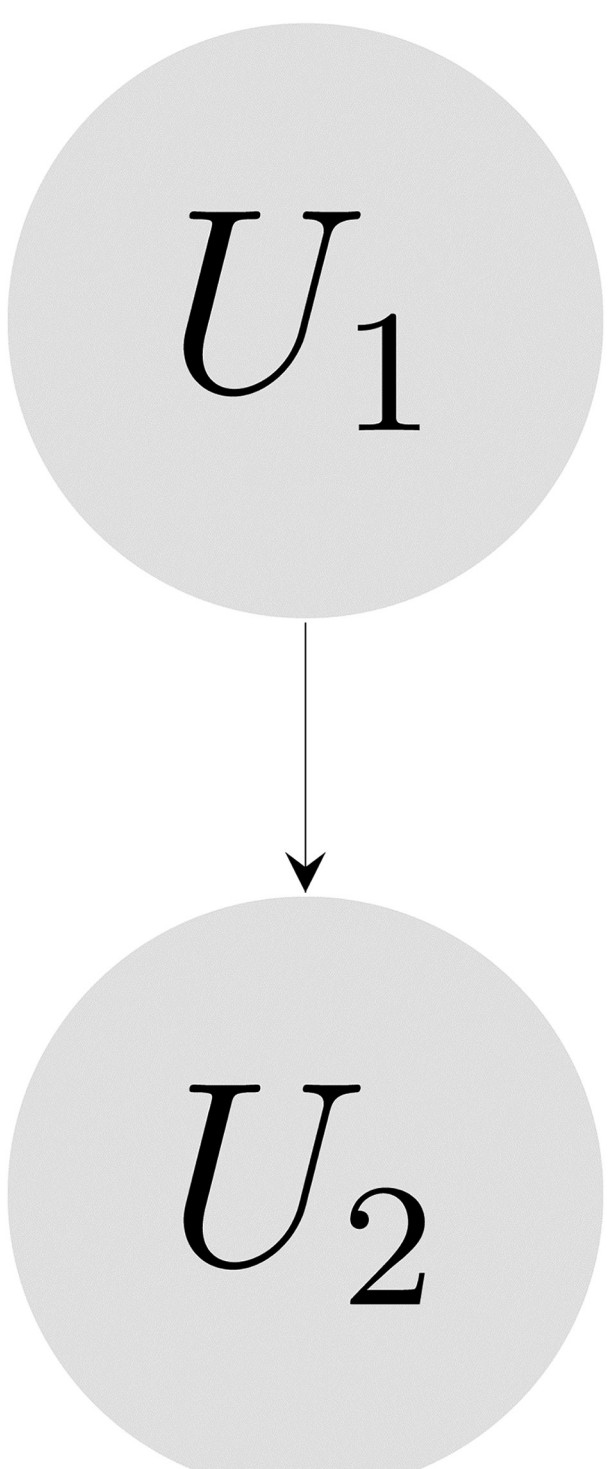

**Fig 1. The graph of the ID wherein $U_1$ influences $U_2$ and both of these nodes are stochastic (indicated by circles).**

Let an *in-out pair* consist of an in-combination—out-combination pair. Let $T$ be the number of time points at which out-combinations are observed, and $\{i_1, \ldots, i_{m_O}\}$ $(m_O \leq m)$ be the set of indices of those group submodels for which at least one out-combination is observed over the observation time interval: $[t_1, t_T]$.

Each of the $e$ output nodes of the ecosystem submodel is stochastic and corresponds to an observable ecosystem metric. A run of the simulator produces a set of simulated values on each output node at each time point. The mean of these values is an estimate of that node's expected value at that time point.

Let $g_s(\mathcal{B}) \in (0, 1)$ be a *goodness-of-fit* statistic that measures the agreement of a sequence of out-combinations and/or mean values of ecosystem metrics produced by a simulator and those of a political-ecological actions data set, $S$ of observed output actions and/or observations on the ecosystem submodel's metrics. Larger values of $g_s(\mathcal{B})$ indicate better agreement. Let $g_H(\mathcal{B}) \in (0, 1)$ be a measure of agreement between the probability distribution on the model's vector of output nodes that is identified by $\mathcal{B}$, and the one identified by $\mathcal{B}_H$. Again, larger values of $g_H(\mathcal{B})$ indicate better agreement. Note that $g_s(\mathcal{B})$ is the agreement between a sample and a stochastic model, while $g_H(\boldsymbol{\beta})$ is the agreement between two stochastic models.

A consistency analysis is executed with the following four steps.

1. **Specify** the values for $\mathcal{B}_H$.

2. **Initialize** the model's parameter values by modifying $\mathcal{B}_H$ to form $\mathcal{B}_{initial}$.

3. **Maximize** the agreement function, $g_{CA}(\mathcal{B})$ by modifying the values of $\mathcal{B}_{initial}$ to form the vector of *consistent* parameter values, $\mathcal{B}_C$.

4. **Analyze** the differences in parameter values between those in $\mathcal{B}_H$, and those in $\mathcal{B}_C$.

The estimator's name comes from this final step: analyze the model's parameters by scrutinizing areas of the subject matter theory that had been used to justify those hypothesis parameter values that, surprisingly, have been found to be very different from their consistent values.

The **Maximize** step of consistency analysis consists of solving

$$\mathcal{B}_C = \arg \max_{\mathcal{B}} \{g_{CA}(\mathcal{B})\} \tag{3}$$

where $g_{CA}(\mathcal{B}) \equiv (1 - c_H)g_s(\mathcal{B}) + c_H g_H(\mathcal{B})$, and $c_H \in (0, 1)$ is the ecosystem manager's priority of having the estimated distribution agree with the hypothesis distribution as opposed to agreeing with the empirical distribution. Setting $c_H$ to zero turns consistency analysis into an MSDE. The subjective assignment of $c_H$ in consistency analysis coupled with its role in the solution of (3) is how consistency analysis represents the reliability of the new data.

The agreement between the simulator's hypothesis distributions and the distributions defined by $\mathcal{B}$ is $g_H(\mathcal{B}) \equiv \frac{1}{m} \sum_{i=1}^{m} g_H^{(i)}(\mathcal{B})$ where

$$g_H^{(i)}(\mathcal{B}) \equiv 1 - \frac{1}{l_i} \sum_{j=1}^{l_i} \hat{\Delta}\left(\beta^{(ij)}, \beta_H^{(ij)}\right), \tag{4}$$

and the estimated Hellinger distance between $\mathbf{U}|\boldsymbol{\beta}_H$ and $\mathbf{U}|\boldsymbol{\beta}$ is

$$\hat{\Delta}(\beta, \beta_H) \equiv \frac{1}{\sqrt{2}} \left[ \sum_{j=1}^{n} \left[ \sqrt{\widehat{pf}_{\mathbf{U}|\beta_H}(\mathbf{u}_j)} - \sqrt{\widehat{pf}_{\mathbf{U}|\beta}(\mathbf{u}_j)} \right]^2 \right]^{1/2}. \tag{5}$$

In this estimator, values of the PDPF under an ID's hypothesis distribution, $\mathbf{U}|\boldsymbol{\beta}_H$ and its $\mathbf{U}|\boldsymbol{\beta}$ distribution are approximated by first drawing a size-$n$ sample of design points from a

multivariate uniform distribution on the ID's chance nodes: $\mathbf{u}_1, \ldots, \mathbf{u}_n$; and then computing $\widehat{pf}_{\mathbf{U}|\boldsymbol{\beta}}(\mathbf{u}_i)$, $i = 1, \ldots, n$ with a nonparametric density estimator.

The agreement between observed output actions and those generated by the simulator is

$$g_S^{(Grp)}(\mathcal{B}) \equiv \frac{1}{m_O T} \sum_{k=1}^{m_O} \sum_{j=1}^{T} I_{\{d_{i_k j} = y_{i_k j}\}}(d_{i_k j}) \tag{6}$$

where $y_{ik}j$ is the observed action of group $i_k$ at time $j$, and $d_{ik}j$ is the submodel-computed action of group $i_k$ at time $j$. Let $S_i \equiv \{z_{i1}, \ldots, z_{iT}\}$ be the $T$ observations on the $i^{\text{th}}$ ecosystem metric. The agreement between observed outputs of the ecosystem and those generated by the ecosystem submodel is

$$g_S^{(Eco)}(\mathcal{B}) \equiv 1 - \frac{1}{eT} \sum_{i=1}^{e} \sum_{j=1}^{T} \frac{|z_{ij} - \hat{z}_{ij}|}{R_i} \tag{7}$$

where $R_i \equiv \max(S_i) - \min(S_i)$. These latter two agreement functions form the overall data agreement function: $g_S(\mathcal{B}) \equiv \left[ g_S^{(Grp)}(\mathcal{B}) + g_S^{(Eco)}(\mathcal{B}) \right]/2$.

## Delete-$d$ jackknife confidence intervals

The deterministic sensitivity analysis described in the next section assumes that confidence intervals for each parameter in $\mathcal{B}$ are available. One way to find these is to compute *delete-d jackknife confidence intervals* (see [41]). Haas [42] gives an algorithm for computing a delete-$d$ jackknife confidence interval. This algorithm proceeds as follows.

1. Resample $r = n^{0.97}$ observations from the observed size-$n$ sample. In other words, temporarily delete $d \equiv n - r$ observations from the observed sample.

2. With this size-$r$ subsample, compute $\beta_1^*$, the consistency analysis estimate of the parameter, $\beta$.

3. Repeat Steps 1 and 2 $n_{jack}$ times to obtain $\beta_1^*, \ldots, \beta_{n_{jack}}^*$.

4. Form a $100(1 - \alpha)\%$ confidence interval for $\beta$ by finding the shortest interval that contains $(1 - \alpha)n_{jack}$ of these $\beta_i^*$ values.

Confidence intervals based on delete-$d$ subsamples are consistent if, as $r \to \infty$, $r/n \to 0$ [43]. One way to meet this condition is to have $r = n^\tau$ where $\tau \in (0, Â 1)$.

## Prediction error rates

The simulator's group submodels produce nominally-valued output in the form of out-combinations. The ecosystem submodel on the other hand, can produce continuously-valued output, e.g. wildlife abundance values. Two different measures of prediction error rate then, are needed. Here, these are the *predicted actions error rate* ($\zeta$) for action-target output, and the *root mean squared prediction error rate* ($\epsilon_i$) for the $i^{th}$ continuously-valued ecosystem metric [8, pp. 186-188].

**Predicted actions error rate.** Consider a finite number of sequential time points, $t_1, \ldots, t_T$. At each of these time points, one or more of the simulator's group submodels posts one or

more out-combinations. Let

$$\zeta \equiv 1 - \frac{1}{T-1} \sum_{i=1}^{T-1} \frac{n_{i+1}^{(match)}}{n_{i+1}^{(obs)}} \tag{8}$$

where $n_{i+1}^{(match)}$ is the number of simulator-predicted out-combinations at time point $t_{i+1}$ that match observed out-combinations at that time point, and $n_{i+1}^{(obs)}$ is the number of these observed out-combinations. It is assumed that the simulator's parameters have been refitted to the political-ecological actions data set using data observed earlier than time point $t_{i+1}$. The justification for this assumption is that an ecosystem manager would want to refit the simulator as new actions and/or values on ecosystem metrics are observed before using the simulator to predict future group actions and/or future values of ecosystem metrics.

Say that a group submodel has $K$ possible out-combinations. In the worst case, one of these out-combinations has a high probability of being chosen at each time point no matter what the input action is. Blind guessing would predict this out-combination with probability $1/K$ at each time point resulting in an error rate of about $1 - 1/K$. An ecosystem manager would prefer the simulator's predictions over predictions based on blind guessing whenever $\zeta < 1 - 1/K$.

**Root mean squared prediction error rate.**   Let

$$\epsilon_i \equiv \left[ \frac{1}{T-1} \sum_{j=1}^{T-1} (z_{i,j+1}^{(obs)} - z_{i,j+1}^{(pred)})^2 \right]^{1/2} \tag{9}$$

where $z_{i,j+1}^{(obs)}$ is the observed value of the $i^{th}$ continuously-valued ecosystem metric at time point $t_{j+1}$, and $z_{i,j+1}^{(pred)}$ is the simulator's predicted value of this metric at time point $t_{j+1}$ where the ecosystem submodel has been fitted to data earlier than time point $t_{j+1}$. Define an alternative predictor, namely the *naive forecast* to be $z_{i,j+1}^{(N)} \equiv z_{i,j}^{(obs)}$. Let $\delta_i$ be the RMSE of these naive forecasts.

**Error rate estimation.**   To estimate these error rates, begin at time point $t_s$, $s > 0$. Then, perform the following two computations at each of the time points $t_s$, $t_{s+v}$, $t_{s+2v}$, ..., $t_j$, ..., $t_{n_{pred}}$ where $v > 0$ is the *refit interval*, $n_{pred} \equiv \lfloor (T_D - 1 - s)/v \rfloor + 1$, $t_{n_{pred}} < T_D$, and $T_D$ is the most recent time point in the data set.

1. Re-fit the simulator with consistency analysis using all observed out-combinations up through time $t_j$.

2. Run this refitted simulator from the first time point in the data set up through time point $t_{j+1}$ to compute predicted values of all output nodes.

With these predictions in-hand, compute an estimate of $\zeta$ with

$$\hat{\zeta} \equiv \frac{1}{n_{pred}} \sum_{j=s}^{n_{pred}} 1 - \frac{n_{j+1}^{(match)}}{n_{j+1}^{(obs)}}. \tag{10}$$

Estimate $\epsilon_i$, and $\delta_i$ with

$$\hat{\epsilon}_i \equiv \left[ \frac{1}{n_{pred}} \sum_{j=s}^{n_{pred}} (z_{i,j}^{(obs)} - z_{i,j}^{(pred)})^2 \right]^{1/2}, \tag{11}$$

and

$$\hat{\delta}_i \equiv \left[ \frac{1}{n_{pred}} \sum_{j=s}^{n_{pred}} (z_{i,j}^{(obs)} - z_{i,j}^{(N)})^2 \right]^{1/2}, \qquad (12)$$

respectively.

Note that the simulator is refitted every $v$ time units. Typically, time is measured in years. An ecosystem manager would be constrained by analyst time, computer availability, and data acquisition frequency. A typical refit time interval is quarterly, i.e., $v = (4 \times 3)/52 = 0.2308$.

If $\hat{\epsilon}_i$ is greater than $\hat{\delta}_i$, the naive forecast is preferred over the model's predictions. In this case, the ecosystem manager would be advised to work on refining and/or modifying the model until $\hat{\epsilon}_i$ is less than $\hat{\delta}_i$.

## Deterministic sensitivity analysis

*Deterministic* sensitivity analysis assesses the sensitivity of a model's outputs to externally-generated values of the model's inputs (see [44]). Haas [8, pp. 182-183] gives an algorithm for studying a simulator's deterministic sensitivity. A new version of this algorithm follows.

**Conditions and responses.** Input for this algorithm consists of a set of *DSA conditions*, $\mathbf{c}_{DSA}$, and a set of *DSA responses*, $\mathbf{r}_{DSA}$. Each of these sets contains values on simulator submodel output nodes. These values can be those of nominally-valued output action nodes, or of continuously-valued ecosystem submodel nodes. Refer to any actions in either of these sets that are to not happen as *complement actions*. A particular pair of these sets embodies a counter-example to the types of simulator outputs that the ecosystem manager is hoping to achieve. Typically, a critic or skeptic of the simulator would specify these sets.

**Algorithm.**

1. Update $\mathcal{B}_H$ to the most recent value of $\mathcal{B}_C$.

2. Specify $\mathbf{c}_{DSA}$, and $\mathbf{r}_{DSA}$ and set the simulator's time interval accordingly.

3. Place all actions contained in either $\mathbf{c}_{DSA}$ or $\mathbf{r}_{DSA}$ into a file of "observed" actions, and all ecosystem responses contained in $\mathbf{r}_{DSA}$ into a file of "observed" ecosystem outputs.

4. Initialize $\mathcal{B}^{(Grp)}$ so that the simulator produces all actions contained in $\mathbf{c}_{DSA}$ and $\mathbf{r}_{DSA}$ but does not produce any complement actions contained in these sets.

5. After setting $c_H$ to 0.1, solve for $\mathcal{B}_{DSA}$ by performing the consistency analysis **Maximize** step (see (3)) using the two files formed in Step 3.

6. Compute $l = \arg \min_{\beta^{(i)} \in \mathcal{B}} |\beta_H^{(i)} - \beta_{DSA}^{(i)}|$.

**Interpretation.** The parameter $\beta^{(l)}$ is the most sensitive parameter, and the difference, $|\beta_H^{(l)} - \beta_{DSA}^{(l)}|$ is the accuracy to which this parameter needs to be known. If $\beta_{DSA}^{(l)}$ is inside the 95% confidence interval for $\beta^{(l)}$ (see the EMT procedure, Step 7), or $\beta_{DSA}^{(l)}$ is a scientifically plausible value for $\beta^{(l)}$, conclude that this analysis supports the skeptic's concerns about the simulator's sensitivity to parameter misspecification.

The idea of this algorithm is to search for a set of parameter values that is as close to $\mathcal{B}_H$ as possible but causes the simulator's outputs to change by an amount that is scientifically significant. If the values in $\mathcal{B}_{DSA}$ are not statistically different from their consistent counterparts or, are scientifically plausible, then the model's outputs are *excessively sensitive* to parameter

misspecification. This sensitivity in-turn, reduces the credibility of policy recommendations derived from the model's outputs.

### Ecosystem management policymaking

Computing the MPEMP is one way to construct an ecosystem management policy. The algorithm described herein is new. Its development was motivated by earlier algorithms given in [8, pp. 52-53], and [17, S5 Appendix]. The idea is to find a set of minimal changes in the beliefs held by ecosystem-affecting groups (relative to their $\mathcal{B}_H^{(Grp)}$ values) so that these groups change their behaviors enough to cause the ecosystem to respond in a desired manner. In other words, the MPEMP is the ecosystem management policy that emerges by finding group submodel parameter values that bring the predicted ecosystem state close to the desired ecosystem state while deviating minimally from $\mathcal{B}_H^{(Grp)}$.

**Definitions.** Let $\mathbf{Q}(\mathcal{B})$ be a random vector composed of a number of the simulator's ecosystem metrics. For example, $\mathbf{Q}(.)$ might consist of cheetah abundance, and herbivore abundance in the year 2030. Assume that an ecosystem manager desires the ecosystem to be in a particular state at a designated future time point. This manager expresses this desired state by specifying the value of $\mathbf{q}_d \equiv E[\mathbf{Q}(\mathcal{B})]$. For example, say that it is desired to have 10,000 herbivores and 1,000 cheetah in East Africa in the year 2030. Then

$$\mathbf{q}_d = (\text{Herbivores} = 10000, \ \text{Cheetahs} = 1000)'. \tag{13}$$

Next, identify those actions that, if taken, would contribute the most towards the ecosystem submodel producing the values in $\mathbf{q}_d$. And, identify those actions that, if ceased, would raise the likelihood of the ecosystem submodel producing the values in $\mathbf{q}_d$. Collect all of these desirable and undesirable actions into a set called $\mathbf{c}_{MPEMP}$. For example, to achieve these desired values, it is believed that more land should be set aside for wildlife reserves, and poaching should cease. In this case,

$$\mathbf{c}_{MPEMP} =$$
$$\left\{ \text{action}^{(\text{kep})} = \left\{ \text{create a new national park} \right\}, \right.$$
$$\left. \text{action}^{(\text{krr})} = \left\{ \text{poach for food, poach for cash, poach for protection} \right\}^C \right\}. \tag{14}$$

where kep, and krr are the Kenya environmental protection agency, and Kenya rural residents groups, respectively.

**MPEMP algorithm.**

1. Update $\mathcal{B}_H$ to the most recent $\mathcal{B}_C$.

2. Compute $\mathbf{q}_H \equiv E[\mathbf{Q}(\mathcal{B}_H)]$.

3. Specify $\mathbf{q}_d$ and $\mathbf{c}_{MPEMP}$.

4. Compute initial values for $\mathcal{B}^{(Grp)}$ with the **Initialize** algorithm of consistency analysis (see Materials and methods: **Consistency analysis**).

5. Compute

$$\mathcal{B}_{\text{MPEMP}} = \arg\max_{\mathcal{B}^{(Grp)}} \left\{ g_H\left(\mathcal{B}^{(Grp)}\right) - \frac{||E[\mathbf{Q}(\mathcal{B})] - \mathbf{q}_d||}{||\mathbf{q}_H - \mathbf{q}_d||} \right\} \tag{15}$$

under the set of constraints specified by $\mathbf{c}_{MPEMP}$.

This algorithm implements one way to quantify the concept of a practical ecosystem management policy: Associate political feasibility with the value of $g_H(\mathcal{B}_{\mathrm{MPEMP}}^{(Grp)})$ where $\mathcal{B}_{\mathrm{MPEMP}}^{(Grp)}$ contains the parameters of the decision making submodels whose values have been modified from those in $\mathcal{B}_H^{(Grp)}$ in such a way that now, the sequence of output actions taken by different groups cause a desired ecosystem state at a designated future time point.

A measure of a plan's political feasibility can be defined as

$$\psi \equiv g_H^{(Grp)}(\mathcal{B}_{\mathrm{MPEMP}})/g_H^{(Grp)}(\mathcal{B}_H). \tag{16}$$

A plan having a value of $\psi$ close to 0.0 will face significant political resistance to its implementation because significant changes to the belief systems of one or more groups needs to happen, while one with a value close to 1.0 should not face such stiff resistance.

## Coding simulator jobs as MTC applications

These five simulator jobs can be computationally expensive. These jobs can, however, be partially parallelized by breaking each of them into sets of dependent tasks that engage in various amounts of data transfer between themselves. Such a set of complex, inter-dependent tasks fits the definition of an MTC application. One way to execute MTC applications is to run them on cluster computers [24, 45]. A cluster computer consists of a number of personal computers called *compute nodes* (hereafter, *nodes*) that are connected through high speed interconnects.

Translating the mathematical expressions of **Materials and methods: Statistical estimation of simulator parameters** into a programming language is performed by writing code within an API that supports the development of task-based parallel programs. A *runtime system* is invoked to execute such programs on hardware. The authors of [46] review APIs and runtime systems that are designed to support MTC applications. These authors refer to a particular combination of an API and a runtime system as a *task-based parallelism technology*.

As identified in [46], an ideal API should be able to direct the runtime system to partition, synchronize, and cancel tasks; specify nodes for workers to run on; start/stop workers; receive task or process fault information; checkpoint a job should a nonrecoverable fault occur; and automatically distribute data and code to workers. In addition, the present author believes that in order to bring many-task computing within reach of ecosystem managers possessing only minimal programming skill, the API should be easy to learn, and use operators whose syntax and semantics are independent of specific runtime systems and hardware configurations.

Therefore, to enable ecosystem managers with different backgrounds to use the five simulator jobs advocated in this article, a task-based parallelism technology needs to possess the following characteristics:

1. Exhibit a high level of abstraction.

2. Be easy to learn.

3. Support the asynchronous, high-level coordination of simultaneous tasks.

4. Separate the communication protocol from the application code.

5. Be internet-aware.

6. Be fault-tolerant: Processor failure is almost certain during a job that employs thousands of processors [47]. Such tolerance implies the ability to automatically checkpoint a job.

7. Be scalable: Only one code need be written and maintained to run jobs on hardware ranging from laptop computers to cluster computers.

8. Be computationally fast.

9. Possess a strong theoretical foundation in computer science.

Currently, several technologies possess some number of these desired characteristics including Java with JavaSpaces, Python with Parsl, Python with Ray, various languages with Docker Swarm, and julia with Docker and Kubernetes. The five simulator jobs could be coded and run in any of these technologies. In what follows, these five technolgies are described and compared.

**Java with JavaSpaces.** The JavaSpaces API can support the *master-worker architecture* wherein a master program runs on one node having a unique Internet Protocol address along with $n_W$ workers who run on other, internet-accessible nodes and busy themselves by executing tasks that have been posted by the master on a JavaSpace bulletin board [48]. One coordination protocol for task posting and collection is the *bag of tasks* scheme wherein the master posts a batch of tasks and then waits until all of these tasks have been completed before posting another batch. This approach results in a program that is naturally balanced and naturally scalable [49]. Noble and Zlateva [50] find that "The simplicity and clean semantics of tuplespaces allow natural expressions of problems awkward or difficult to parallelize in other models [51]." A JavaSpaces program is also fault tolerant and decouples the semantics of distributed computing from those of the problem domain [49].

The runtime system Gigaspaces™ that supports the JavaSpaces API exhibits low inter-node communication latency [52]. The primary operations on a Gigaspaces space are `write`, `read`, `change`, `take`, and `aggregation` [53, 54]. Appendix A of S1 Appendix contains shell scripts that start and run a JavaSpaces program on a cluster computer. Appendix B of S1 Appendix contains guidance for running a JavaSpaces program on a shared cluster computer.

**Python with Parsl.** The Parsl package allows distributed Python programs to access thousands of nodes [55] either on cluster computers or in the cloud. The distributed application is created using the API operators `Config`, `@python_app`, and `@bash_app`.

**Python with Ray.** The Python package, *Ray* [56] provides the API operators `@ray.remote`, `ray.wait`, `ray.get`, and `ray.put`. Ray contains it own runtime system to manage the starting, reading, deleting, and recovery of tasks [57].

**Various languages with Docker and Docker Swarm.** Docker is a program that takes application language source code and creates a portable and executable version called a *container*. Docker Swarm Mode is a runtime system that orchestrates the execution of these containers across nodes on a cluster computer or in the cloud. Docker Swarm Mode can be used to manage a task-based, multi-language distributed program [58]. The steps needed to do this are 1) write the application modules in various application languages, 2) start support programs on each node, 3) start a Docker Swarm cluster by executing commands on each node, 4) create a Docker registry, 5) create images and from them, containers, 6) register the images, 7) create a stack file, and 8) run the application by deploying this stack.

**julia with Docker and Kubernetes.** The *julia* language [59] contains an API that provides the `@spawn`, and `fetch()` operators needed to run a bag-of-tasks application [59]. To do this, one needs to first use Docker to containerize the julia-written executables. Then, these containers are run on a Kubernetes cluster [60].

**Comparisons.** All five technologies are known to coordinate tasks, be internet-aware, and be computationally fast. Table 1 summarizes the strengths and weaknesses of these five technologies. Two notes are in order. JavaSpaces has a theoretical foundation in computer science [51, 52] that the other four technologies lack. Developing an MTC application with Docker Swarm Mode appears to require more user involvement with the runtime system than the other four technologies. On the other hand, container-based software development and distribution is quickly becoming the industry standard.

**Table 1. Comparison of task-based technologies on desirable characteristics for building and running MTC applications.**

| Characteristic | JavaSpaces | Parsl | Ray | Docker Swarm | julia |
|---|---|---|---|---|---|
| Abstraction | known | known | known | NA | known |
| Easy to learn | known | known | known | NC | known |
| Communication hiding | known | known | known | NA | known |
| Scalable | known | known | known | known | known |
| Fault-tolerant | known | known | known | known | NC |
| Portable | known | NC | NC | known | NC |
| Strong theoretical foundation | known | NC | NC | NC | NC |

Not Clear (NC) indicates a document verifying the characteristic could not be found. Known means the technology is known to possess the characteristic either from publication or computational experience. Because Docker Swarm is a runtime system, Abstraction and Communication hiding are not applicable (NA) to it.

This author chose the JavaSpaces API to develop the MTC applications exercised in the next section rather than any of the other four technologies because it is the only technology known to possess all of the desirable characteristics listed in Table 1.

**Optimization as an MTC application.** Optimization of stochastic functions under non-linear constraints can be performed with the *multiple dimensions ahead search* (MDAS) algorithm of Haas [8, pp. 219-225]. This algorithm is a parallel version of the Hooke and Jeeves coordinate search algorithm [61]. MDAS executes by having the master assign each worker a vector of parameter values at which to compute the value of the objective function. These vectors are chosen such that the next $M$ parameters are searched simultaneously for a maximum. Each worker computes the objective function value at its assigned set of parameter values. Once all of the workers have returned their function values, the master checks them for a new maximum (called an *improvement*). If found, the master stores this new best solution. This parallel search is repeated on these dimensions until no improvements are found. Then, the algorithm moves on to the next $M$ dimensions.

This algorithm was benchmarked against the classic Bukin F4 function [62]:

$$f(x, y) = 100y^2 + .01|x + 10| \tag{17}$$

for $x \in [-15, -5]$, and $y \in [-3, 3]$. Starting at $(-6, 2)$, MDAS found the global minimum of zero at the point $(-10, 0)$ after 1081 function evaluations.

## Simulator job-specific algorithms and runtime issues

Algorithmic details for how each simulator job is converted to an MTC application follow.

**Consistency analysis.** Consistency analysis is run as an MTC application by performing its **Maximize** step with the MDAS algorithm wherein each worker runs on its own node. In order to both speedup evaluation of the objective function and to improve the optimization run's convergence behavior, smooth objective functions are employed in-lieu of those based on the approximate negative Hellinger distance for $g_H^{(Grp)}$, and $g_H^{(Eco)}$ (see (4)). These functions are the negative of the Euclidean distance between the parameters at their hypothesis values and those at a particular trial point in the optimization run. Call these Euclidean agreement measures $e_H^{(Grp)}$, and $e_H^{(Eco)}$, respectively.

**Credibility assessment and the MPEMP.** Jackknifing involves executing consistency analysis on each of $n_{jack}$ separate delete-$d$ subsamples. It can be implemented as an MTC application by performing all of these $n_{jack}$ consistency analysis tasks simultaneously.

Converting the prediction error rate job to an MTC application involves running a consistency analysis task on each of $n_{pred}$ subsamples (see Materials and methods: **Prediction error rates**). This is accomplished the same way that the jackknife subsamples are processed.

The computational demands of a deterministic sensitivity analysis accrue from the consistency analysis performed in its Step 3 (see Materials and methods: **Deterministic sensitivity analysis**).

The computational demands of the MPEMP job accrue from the optimization problem solved in the MPEMP algorithm's Step 5 (see Materials and methods: **Ecosystem management policymaking**). This job is implemented in a way similar to consistency analysis.

## Case study description

The following Results section contains a case study that applies the five simulator jobs to the estimation, credibility assessment, and MPEMP computation of an EMT for the conservation of cheetah in East Africa. All input files for this simulator are available at [63]. Hereafter, this simulator is referred to as the *cheetah EMT simulator*.

**Overview of the cheetah EMT simulator.** Haas [8] builds a simulator of the interactions between cheetah and humans in the East African countries of Kenya, Tanzania, and Uganda. The model consists of group submodels for each country's presidential office (kpr, tpr, upr), environmental/wildlife protection agency (kep, tep, uep), non-pastoralist, rural residents (krr, trr, urr), and pastoralists (kpa, tpa, upa). In addition, a submodel is built to represent the group of conservation NGOs who have operations in at least one of these countries (ngo). All of these group submodels can interact with each other. And, each country's environmental protection agency, rural residents, and pastoralists submodels can directly interact with a submodel of the ecosystem that spans these three countries (ecosys). This ecosystem hosts populations of cheetah and their herbivore prey. This model is formally documented in Appendix C of (S1 Appendix).

An automatic data acquisition system has been gathering data since January, 2007 on this political-ecological system (see [20]). This data set contains 1555 actions observed from the year 2002 to 2019. S1 Data contains this data set. A portion of this data reveals a complex pattern of group actions followed by reactions from other groups (Fig 2). Cheetah abundance data is taken from [64, 65], and [66].

## Results

### Consistency analysis

Consistency analysis was used to estimate the parameters of the node: scenario imminent interaction with police within the Kenyan rural residents group submodel. A time step of 13 days results in each time interval containing about five actions. The **Initialize** step of consistency analysis was run to produce a set of initial parameter values. The initial match fraction (the ratio of the number of observed actions matched by the simulator's output to the number of observed actions) is 0.646. The fraction of actions matched regardless of whether the target was matched, is 0.772, and the corresponding target match fraction is 0.870. See Table 2 for individual submodel match fractions.

Next, the **Maximize** step of consistency analysis was run on the Triton Shared Computing Cluster (TSCC) at the San Diego Supercomputer Center [67]. For this run, $c_H$ was set to 0.99, and each belief network was simulated with 1000 Monte Carlo realizations. Nine nodes were employed and the maximum number of function evaluations was set to 1200. Only those parameters having an initial value different from their hypothesis value were modified. This resulted in only 40 of the 459 parameters being active during the optimization run—a significant reduction in the problem's dimensionality. Initial and final values under the stochastic

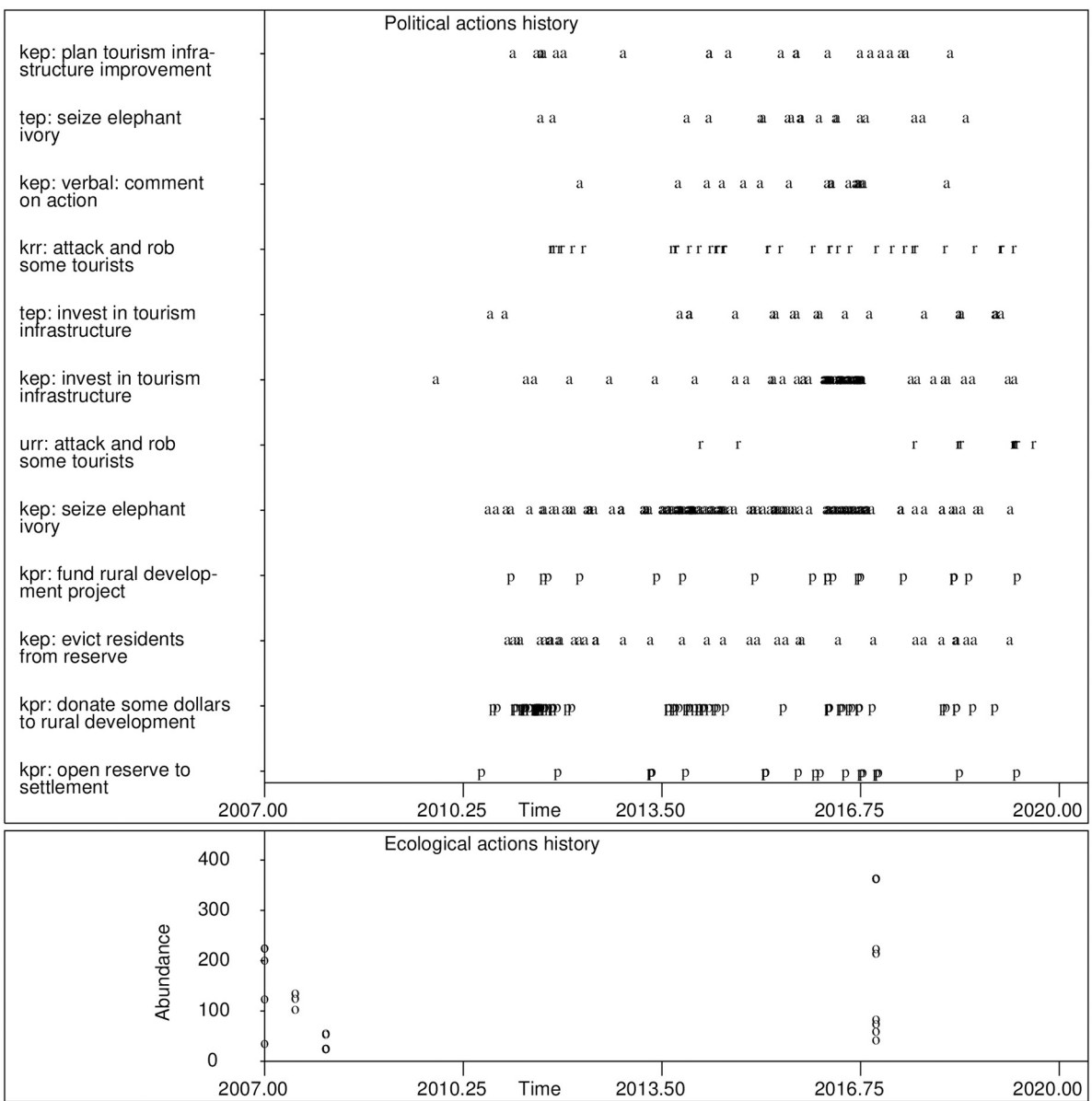

**Fig 2. Observed actions history from East African online news stories for the period from January 2007 through June 2019.** The symbol "p" indicates an action taken by a presidential office, "a" an action taken by an EPA, "r" an action taken by rural residents, "s" an action taken by pastoralists, and "n" an action taken by an NGO. Selected out-combinations only are labeled. The bottom plot is observed cheetah abundance.

agreement measure for $g_H(.)$ (4) were computed using 5000 Monte Carlo realizations for each belief network.

Under this configuration, the simulator job's wall clock time was 4.42 hours. The solution achieved a 25.5% increase in $g_{CA}(\mathcal{B})$ (Table 3).

## Delete-*d* jackknife confidence intervals

Jackknife confidence intervals were computed for the parameters that define the `scenario imminent interaction with police` node in the Kenya rural residents submodel of the cheetah EMT simulator. The jackknife subsample size is $r = 546^{0.97} = 451$, and $n_{jack} = 5$.

**Table 2. Match fractions from the initialize step of consistency analysis for the cheetah EMT simulator.**

| Submodel | $n_{obs}$ | $n_{match}$ | Match fraction | $n_{actmatch}$ | Action match fraction | $n_{trgtmatch}$ | Target match fraction |
|---|---|---|---|---|---|---|---|
| kpr | 1 | 0 | 0 | 0 | 0 | 0 | 0 |
| kep | 142 | 90 | 0.633 | 90 | 0.633 | 141 | 0.992 |
| krr | 1 | 0 | 0 | 0 | 0 | 1 | 1.000 |
| kpa | 0 | 0 | 0 | 0 | 0 | 0 | 0 |
| tpr | 0 | 0 | 0 | 0 | 0 | 0 | 0 |
| tep | 27 | 15 | 0.555 | 15 | 0.555 | 27 | 1.000 |
| trr | 0 | 0 | 0 | 0 | 0 | 0 | 0 |
| tpa | 0 | 0 | 0 | 0 | 0 | 0 | 0 |
| upr | 0 | 0 | 0 | 0 | 0 | 0 | 0 |
| uep | 24 | 15 | 0.625 | 15 | 0.625 | 24 | 1.000 |
| urr | 0 | 0 | 0 | 0 | 0 | 0 | 0 |
| upa | 0 | 0 | 0 | 0 | 0 | 0 | 0 |
| ngo | 131 | 90 | 0.687 | 131 | 1.000 | 90 | 0.687 |
| ecosys | 0 | 0 | 0 | 0 | 0 | 0 | 0 |

**Table 3. Consistency analysis agreement measures for the cheetah EMT simulator.**

| Agreement Measure | Initial Value | Final Value |
|---|---|---|
| $g_S^{(Grp)(\mathcal{B})}$ | 0.6308 | 0.6000 |
| $e_H^{(Grp)}(\mathcal{B})$ | -41.6800 | -29.4314 |
| $g_H^{(Grp)}(\mathcal{B})$ | 0.8468 | 0.8888 |
| $g_{CA}(\mathcal{B})$ | -1.1394 | -0.8483 |

These five subsamples were used to compute 50% confidence intervals. Nine nodes ran for 4.85 wall clock hours to complete the job. All parameters are significantly different than zero. The five widest confidence intervals (Table 4) indicate that estimates of the group's beliefs about being prosecuted for actions they might take are not excessively affected by sampling variability.

## Prediction error rates

Prediction error rate was estimated by computing one-step-ahead predictions of actions, and cheetah abundance from 2016.9 through 2018. This run required 3.25 wall clock hours on the

**Table 4. The five widest confidence intervals of parameters defining the node `Scenario Imminent Interaction With Police (SIIWP)` in the Kenya rural residents submodel.**

| ECON conditioning value | IIWP conditioning value | Lower boundary | Upper boundary | Width |
|---|---|---|---|---|
| *negligible* | *will be evicted* | 0.110 | 0.362 | 0.252 |
| *negligible* | *will be arrested* | 0.161 | 0.412 | 0.251 |
| *negligible* | *no interaction* | 0.211 | 0.462 | 0.251 |
| *inadequate* | *will be arrested* | 0.111 | 0.262 | 0.151 |
| *adequate* | *will be arrested* | 0.111 | 0.262 | 0.151 |

These parameters are conditional probability values. The conditioning nodes are `scenario action (ACTN)`, `situation economic goal (ECON)`, and `situation imminent interaction with police (IIWP)`. For all five of these intervals, the conditioning value for `ACTN` is *poach for cash*, and the `SIIWP` node's value is *no interaction with police*.

TSCC running nine nodes. The run produced 57 predictions resulting in $\hat{\zeta} = 0.4667$, and $\hat{\epsilon} = 140.0$ for the cheetah abundance metric. The simulator was refitted to data five times.

### Deterministic sensitivity analysis

Say that the ecosystem manager wishes to use the simulator's outputs to justify his/her position that reducing poaching would slow or reverse the decline in cheetah abundance. A skeptic, however, believes that scientifically plausible parameter values in the cheetah submodel can be found such that when the model is run from 2019 through 2025 under the restriction of no poaching actions, cheetah abundance in the year 2025 will be insignificantly different than that produced by the simulator when run under the assumption that current poaching rates continue into the future. If such parameter values can be found, the skeptic would argue that the model is unable to inform management action selection because the model can be calibrated to either recommend increased antipoaching effort or not recommend increased antipoaching effort.

To represent this skeptic's belief, $\mathbf{c}_{DSA}$ consists of the single constraint: *no poaching actions occur from the present through the year 2025*, i.e.,

$$\mathbf{c}_{DSA} = \{\texttt{action}^{(\texttt{krr})} = \{poach\ for\ food,\ poach\ for\ cash,\ poach\ for\ protection\}^{C}\}. \quad (18)$$

And, $\mathbf{r}_{DSA}$ is populated with predictions of expected cheetah abundance in the year 2025 across several regions in Kenya (Table 5). These predicted values are found by running the simulator out to the year 2025 under the consistent parameter values found in **Results: Consistency analysis**. It is the use of these consistent values that forces poaching rates from 2019 through 2025 to be equal to current poaching rates.

The mathematical programming problem (3) with variables consisting of the ecosystem submodel's parameters was solved over the interval 2019 through 2025 and required one hour of wall clock time on the TSCC utilizing eight worker nodes. Initial parameter values were set to $\mathcal{B}_{H}$ with the exception that values in $\boldsymbol{\beta}^{(\texttt{krr})}$ were adjusted as necessary so that any contemplated poaching action produced a small value of $E[\texttt{Overall Goal Attainment}]$. Doing so caused the Kenya rural residents group to avoid poaching actions during the optimization.

If a solution to (3) were found such that all values in $\mathcal{B}_{DSA}$ were scientifically plausible, then the skeptic's position would be supported. As Table 6 indicates, however, the skeptic's position is not supported because the value for the initial death rate, $r_0$ (see Appendix C of S1 Appendix) needed to respect the conditions in $\mathbf{c}_{DSA}$ and the responses in $\mathbf{r}_{DSA}$, is unrealistically high (0.510) under minor poaching pressure.

**Table 5. Cheetah abundance predictions in five regions of Kenya for the year 2025 computed under consistent parameter values.**

| Region | Abundance |
|---|---|
| Laikipia | 200 |
| Samburu | 200 |
| Tsavo | 145 |
| Marsabit | 200 |
| Turkana | 40 |

These values make up the set $\mathbf{r}_{DSA}$.

**Table 6. Results for the deterministic sensitivity analysis of the ecosystem submodel.**

| Parameter | Hypothesis value | DSA value |
|:---:|:---:|:---:|
| | minor poaching pressure | |
| $r_0$ | 0.043 | 0.510 |
| $\alpha_r$ | 0.000 | 0.000 |
| $\beta_r$ | 0.001 | 0.001 |
| | moderate poaching pressure | |
| $r_0$ | 0.400 | 0.220 |
| $\alpha_r$ | 0.000 | 0.000 |
| $\beta_r$ | 0.001 | 0.001 |
| | severe poaching pressure | |
| $r_0$ | 0.600 | 0.600 |
| $\alpha_r$ | 0.010 | 0.010 |
| $\beta_r$ | 0.001 | 0.001 |

## Credibility assessment of the cheetah EMT simulator

The cheetah EMT model's mechanism reflects principles of how political-ecological systems function [8, chs. 6-8]. Hence, component (a) of the Patterson and Whelan [11] criteria (see Introduction) is satisfied. Statistical estimation of the model's parameters is the foundational step for establishing components (b) and (c). The model's confidence intervals indicate that a selection of the model's parameters cannot be ignored and can be estimated without excessive uncertainty. The model's prediction error rates, however, are high. Finally, the model is resistant to a skeptic-created scenario engineered to show the model being unable to inform management action selection.

## Finding the MPEMP

Say that it is desired to have 5,000 herbivores and 500 cheetah in East Africa in the year 2030. These target values are expressed by specifying

$$\mathbf{q}_d = (\texttt{HrbvrNm}(2025) = 3000, \texttt{ChthNm}(2025) = 200,$$
$$\texttt{HrbvrNm}(2030) = 5000, \texttt{ChthNm}(2030) = 500)'. \tag{19}$$

To achieve this ecosystem state, more land needs to be set aside for wildlife reserves, and poaching needs to cease. These conditions are expressed by setting

$$\mathbf{c}_{MPEMP} =$$
$$\left\{\texttt{action}^{(\texttt{kenepa})} = \{create\ a\ new\ national\ park\}\right\},$$
$$\texttt{action}^{(\texttt{kenrr})} = \{poach\ for\ food,\ poach\ for\ cash,\ poach\ for\ protection\}^C\right\}. \tag{20}$$

Group beliefs that are to be changed are those of the `imminent interaction with police` node of the Kenya rural resident group.

The simulator job for finding the MPEMP formed a 108-dimensional optimization problem. When run with eight worker nodes on the TSCC, this simulator job required 2.97 wall clock hours to complete. Initial and final values of $g_H^{(\texttt{krr})}(\mathcal{B})$ (4) were computed using 5,000 Monte Carlo realizations for each belief network. The MPEMP actions history (Fig 3) is such that Kenyan rural residents substitute the action *verbally protest national park boundaries* for

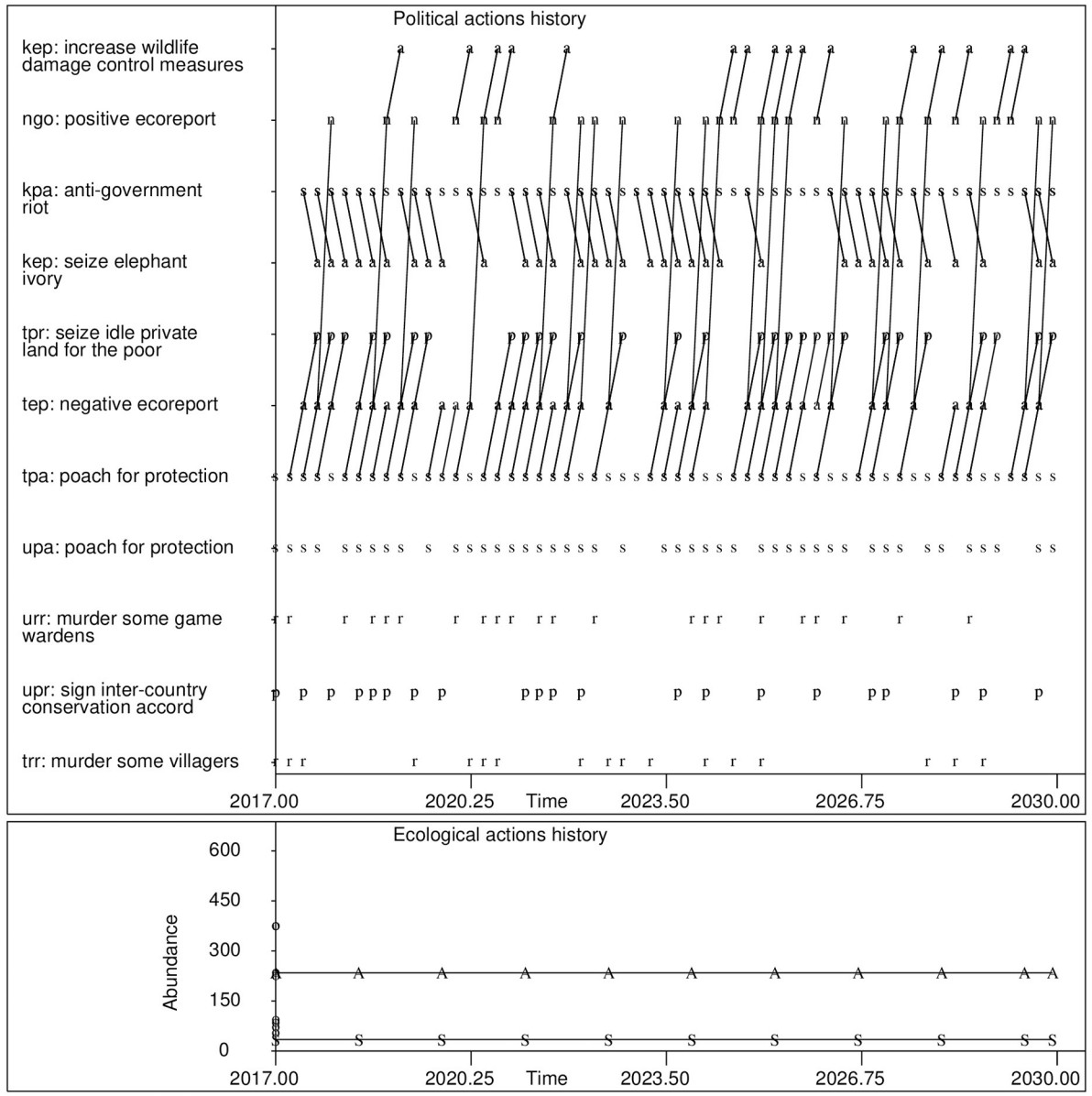

**Fig 3. The cheetah EMT simulator's actions history under the MPEMP.** See Fig 2 for symbol legend. Lines connect action-reaction sequences. For example, one frequent action sequence in Tanzania is poaching, followed by a negative ecosystem status report, followed by a land gift to the poor.

poaching actions. In spite of this behavioral change, however, cheetah abundance does not attain the desired level by the year 2030.

This plan's $\psi$ value is 0.845 meaning that this plan is not expected to face severe resistance to its implementation. This result rests on the nearness of the hypothesis distributions to the MPEMP distributions of the rural residents and pastoralists submodels. These hypothesis distributions represent recent efforts to include local people in the management of protected areas. Abukari and Mwalyosi [68] report that local people will find a protected area advantageous to their livelihoods if they are included as equal participants in decisions concerning the

management of the protected area and, for nonpastoralists, if there is land outside the protected area where they can grow crops.

## Total compute time

In this case study, running the five simulator jobs on a modest number of parameters, required 20 hours of wall clock time using 10 nodes. Due to the "curse of dimensionality," if a larger number of parameters were assessed, this time could increase by two orders of magnitude. Say that the data set is updated quarterly as suggested in **Materials and methods: Prediction error rates**. Then, if these jobs were rerun after every update as called for in Step 11 of the EMT procedure, an ecosystem manager would need 20,000 hours of wall clock time every three months were he/she to run them on a single workstation. Clearly some sort of parallel computing alternative is needed.

## Discussion

A procedure has been described for developing models of political-ecological systems that characterize the dynamics of an ecosystem being impacted by and impacting several different groups of humans. As part of this procedure, an integrated suite of methods has been presented for assessing a model's credibility and computing ecosystem management plans with it. Through a case study, downloadable software [39] has been demonstrated that implements these methods as MTC applications. Doing so is a cost-effective way to support the lengthy computations that these methods entail.

Further computational evidence on these methods is provided by first, the consistency analysis of a rhino conservation simulator reported in [17]. There, the authors fit 145 parameters of the rhino poacher decision making submodel. Second, a deterministic sensitivity analysis is performed on a different rhino conservation simulator in [19] where it is concluded that the model is not excessively sensitive to 10 key parameters.

The data streams used for model estimation need to contain observations on more of the model's outputs in order to establish the credibility of the group decision making submodels. Because of the massive amount of computation called for in this article, more efficient optimization algorithms also need to be developed. Fault recovery needs to be an integral part of these algorithms. Finally, the EMT procedure given herein needs to be used to develop group decision making submodels that learn.

This article provides for the first time, a way for ecosystem managers to develop credible models with which to manage ecosystems that contain endangered species. Given the decline in the earth's biodiversity, the potential impact of this contribution is difficult to overstate. But the future of ecosystem management lies in finding workable policies that not only address what needs to be done to conserve ecosystems under anthropogenic pressure, but also address the needs and aspirations of those people who interact with such ecosystems. Developing credible models of these political-ecological systems via the EMT procedure described herein can make this happen.

## Supporting information

**S1 Appendix. Shell scripts, guidance, and model documentation.** Shell scripts to initiate a Gigaspace, guidance for running on a shared cluster computer, and documentation of the cheetah EMT simulator.
(PDF)

**S1 Data. Observed actions history for the Cheetah EMT simulator.** All data used in the cheetah conservation case study.
(TXT)

## Acknowledgments

I thank two anonymous reviewers for suggestions that improved the manuscript.

## Author Contributions

**Conceptualization:** Timothy Haas.

**Data curation:** Timothy Haas.

**Formal analysis:** Timothy Haas.

**Methodology:** Timothy Haas.

**Software:** Timothy Haas.

**Validation:** Timothy Haas.

**Visualization:** Timothy Haas.

**Writing – original draft:** Timothy Haas.

**Writing – review & editing:** Timothy Haas.

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
