## [Decision Letter · Decision Letter 0]

24 Feb 2020

PONE-D-19-33903

Developing political-ecological theory: The need for Many-Task Computing

PLOS ONE

Dear Professor Haas,

Thank you for submitting your manuscript to PLOS ONE. After careful consideration, we feel that it has merit but does not fully meet PLOS ONE’s publication criteria as it currently stands. Therefore, we invite you to submit a revised version of the manuscript that addresses the points raised during the review process.

We would appreciate receiving your revised manuscript by Apr 09 2020 11:59PM. To enhance the reproducibility of your results, we recommend that if applicable you deposit your laboratory protocols in protocols.io, where a protocol can be assigned its own identifier (DOI) such that it can be cited independently in the future. For instructions see: http://journals.plos.org/plosone/s/submission-guidelines#loc-laboratory-protocols

We look forward to receiving your revised manuscript.

Kind regards,

Christian Vincenot, Ph.D.

Academic Editor

PLOS ONE

Additional Editor Comments (if provided):

The reviewer raises several concerns regarding your study. These should all be possible to address, so I return the manuscript to you with major revisions. FYI, I intend to gather the opinion of a second reviewer from ecology in the second round to also assess the conservation part of your work.

Journal Requirements:

Reviewers' comments:

Reviewer's Responses to Questions

**Comments to the Author**

1. Is the manuscript technically sound, and do the data support the conclusions?

Reviewer #2: Yes

2. Has the statistical analysis been performed appropriately and rigorously? 

Reviewer #2: Yes

3. Have the authors made all data underlying the findings in their manuscript fully available?

Reviewer #2: Yes

4. Is the manuscript presented in an intelligible fashion and written in standard English?

Reviewer #2: Yes

5. Review Comments to the Author

Reviewer #2: The article is focused on developing political-ecological theory. To this purpose, it considers that models should be better analysed in depth in order to increase their credibility.

This comes with a huge need of High Computing Performance. Authors thus propose to use Many-Task Computing coded with JavaSpacesTM.

The major critics again this article is that it considers many parts (especially in Section 1 and 2), that are related to the subject, but not directly to the contribution of the article.

This makes the article too long and a little bit confused on its main objective.

As an example all the part about agent-based modeling can be omitted (or reduced a lot).

Similarly does the presentation of all the statistical tools really need to be so detailled, as the main question of the paper is on parallelisation of the computation ?

In contrarily, the choice of the Many-Task Computing approach and coded with JavaSpacesTM is not supported by a state of the art of the existing alternatives approaches.

l31: "There is a need to acknowledge the complexity of political-ecological systems"

There is not a need anymore to acknowledge the complexity of political-ecological systems, t

l54 : "Unless such a model is shown to be credible using in-part, appropriate statistical methods, any "recommendations based on output from the model may receive only mixed acceptance by those affected."

This has been showed that, even if the model is well-calibrated and analysed, it can be not been accepted by decision-makers, in particular due to a lack on involvement (see articles of A. Smagl) or that many models are accepted even if not realistic or calibrated (cf. all the works in participatory approaches and participative simulation).

l135: "A literature search uncovered only two articles describing the statistical estimation of a socio-ecological model's parameters, namely, [23], and [17]."

What about: "Calibration of simulation platforms including highly interweaved processes: the MAELIA multi-agent platform. June 2014, Conference: 7th Intl. Congress on Env. Modelling and SoftwareAt: San Diego, CA, USA, Romain Lardy et al. "

In addition, it would be surprising that in all the models of socio-environmental systems (being agent-based or not), there is no parameter estimation...

Finally, the issue of sensitivity analysis, calibration, parameter estimation ... could be tackled more generally than only in the context of SES systems, but also in any other applications (e.g. urban context, cf Denise Pumain & Romain Reuillon. Urban Dynamics and Simulation Models.)

p8: "A political-ecological system simulator (hereafter simulator) is an executable computer program capable of approximating the outputs of a stochastic model of a political-ecological system."

This definition can be applied in many cases of models. But in the case of agent-based models (that is the kind of models that is presented in this paper), defining it only related to its outputs is limitative, as its main strength relies on its capability to represent and understand inner dynamics. To fit with outputs, modern AI models are often more accurate.

l215: this is only a technical point of view that occults all the social acceptability and power game that are important in model adoption.

l228: "EMT proceudre"

- how is it related to the idea of modeling methodolofy ?

- Step 1 : what does the idea of boundaries cover ? is it spatial, temporal ? thematic ?

- Do we need to specialize an ecosystems to one with endanger species ? Why does this approach does not aim at more generality ?

- the various steps of the procedure have very different level of precision: there are details about files on one side (very low and technical level) and other parts are really higher level (e.g. identify the ecosystem ... )

As parallelisation seems to be the key point of the paper a State of the Art about parallelization technics and MTC would be necessary.

In particular, only JavaNameSpace and MTC are presented and no other alternatives.

l1000 : The conclusion that taking more land to build a reserve with a low resistance to its implementation seems unrealistic.

This would requiers more details.

6. PLOS authors have the option to publish the peer review history of their article (what does this mean?). If published, this will include your full peer review and any attached files.

Reviewer #2: No

---

## [Author Response · Author response to Decision Letter 0]

27 Mar 2020

Please see my Response to Reviewers file.

---

## [Decision Letter · Decision Letter 1]

21 Jul 2020

PONE-D-19-33903R1

Developing political-ecological theory: The need for Many-Task Computing

PLOS ONE

Dear Dr. Haas,

Thank you for submitting your manuscript to PLOS ONE. After careful consideration, we feel that it has merit but does not fully meet PLOS ONE’s publication criteria as it currently stands. Therefore, we invite you to submit a revised version of the manuscript that addresses the points raised during the review process.

In two rounds of review now, very similar comments have been made by several reviewers. In summary, it has been repeatedly recommended that the manuscript be significantly reduced in length, providing a clearer, tighter focus on its contribution, and allowing readers to more easily engage with, and make use of, the ideas presented. 

You are strongly urged to take this advice when addressing the reviewers' comments. The current manuscript, with some appropriate editing, might be suitable for a chapter in a specialist collected volume - it is far too long and discursive for publication in a journal such as PLOS ONE. An appropriate response to this issue will be required for the manuscript to be accepted for publication.

In addition, it is recommended that you take note of, and address, comments regarding the computational validation presented, and the need for a broader consideration of possible approaches to achieving parallel implementation.

We look forward to receiving your revised manuscript.

Kind regards,

Andrew Lewis

Academic Editor

PLOS ONE

Reviewers' comments:

Reviewer's Responses to Questions

**Comments to the Author**

1. If the authors have adequately addressed your comments raised in a previous round of review and you feel that this manuscript is now acceptable for publication, you may indicate that here to bypass the “Comments to the Author” section, enter your conflict of interest statement in the “Confidential to Editor” section, and submit your "Accept" recommendation.

Reviewer #3: (No Response)

Reviewer #4: (No Response)

2. Is the manuscript technically sound, and do the data support the conclusions?

Reviewer #3: Partly

Reviewer #4: Yes

3. Has the statistical analysis been performed appropriately and rigorously? 

Reviewer #3: No

Reviewer #4: Yes

4. Have the authors made all data underlying the findings in their manuscript fully available?

Reviewer #3: No

Reviewer #4: Yes

5. Is the manuscript presented in an intelligible fashion and written in standard English?

Reviewer #3: Yes

Reviewer #4: No

6. Review Comments to the Author

Reviewer #3: In its current form, the manuscript is unpublishable. I say this not unkindly, but constructively. The manuscript reads far more like a thesis, but not one that would be passable. The manuscript is far too long and contains a lot of irrelevant material that obscures its contributions (the important bits). I believe that contributions are there, but they are completely obscured. Some points that will turn that manuscript into a journal paper worthy of publication in PLOS ONE are:

* The emphasis is all wrong. The manuscript is to much front loaded particularly in the introduction and literature review sections. These need to short and sharp and really drive the motivation of the novel contribution of the work. At the moment they are bloated and contain so much discussion that the reader easily forgets what the paper is about. Similarly, the important back-end is very scant. There needs to be a much richer discussion of results, the summary of the work and particularly what the future work can be.

* In terms of the front-end, why do you have all these very large quotes? They add nothing to the paper. Remove them all and just paraphrase if necessary.

* Section 1.3: A new algorithm (step-by-step procedure) is added in the literature review section. This would need to go later in the paper.

* In general, please make algorithms short with minimal explanation each step, otherwise they become difficult to read. If you need more explanation, provide it in the body of the text.

* There is a Figure 2 caption, but no actual figure.

* As intimated above, more extensive computational evidence would be good.

* Try to read some other PLOS ONE papers to see the form of them. I'm sure you'll find this helpful when you are revising your manuscript.

Having said all of that, I'm really looking forward to reviewing a revised version of this manuscript!

Reviewer #4: It's important to note that I was not one of the original reviewers. Given they did a good job of identifying issues of concern I have focused on their comments and the responses to them, rather than reanalysing the manuscript from scratch. With this in mind, this task would have been made significantly easier if the track changes version had been produced used latexdiff (https://ctan.org/pkg/latexdiff) instead of manual bolding of content. While the Comment n markers were helpful in linking sometimes modified, typically additional, content to the relevant comments, it was impossible to see if content was deleted or otherwise edited without going through both versions side-by-side. This showed that some of the bolded content has not been modified while some other changes have been made that have not been marked at all.

It is clear, from the detail in the manuscript and the responses to the original reviewer's comments, that the author is passionate about the work and about making it technically sound. I appreciate the additions to the work have been made to address some of that reviewer's comments. However, the original reviewer clearly identified that the work was too long, with many details that could have been reduced or omitted (because they are covered elsewhere in the literature) and this revised version is even longer, at the upper end of a typical Honours (2-semester research) thesis.

Based on my reading I am compelled to agree. Either the foundational work in Sections 1 and 2 could have been reduced to more tightly focus the paper, or the paper should have been split into two works. If the goal of the work is to provide (and evaluate & justify) a new tool for ecological management, then perhaps it would be better to split the material into (1) a paper that justifies its validity and shows its promise and (2) a suite of technical documents hosted with the open-source software. This would go some way toward dealing with the original reviewer's comments #1 and #11.

Re Comments 6/7: The original reviewer's point remains and could be expanded as "Perhaps the search query was too narrow and missed relevant literature". The responses only address the offered example, not the thoroughness of the original search.

Re original reviewer comment #12, they have a point. There are now many approaches to achieving parallelism. What they left out was a listing of examples that _do_ support easy deployment of applications across heterogeneous devices, like Docker, which are ignored in this paper.

7. PLOS authors have the option to publish the peer review history of their article (what does this mean?). If published, this will include your full peer review and any attached files.

Reviewer #3: No

Reviewer #4: No

---

## [Author Response · Author response to Decision Letter 1]

30 Jul 2020

Please see the uploaded file "r2rspns.pdf.

---

## [Decision Letter · Decision Letter 2]

16 Sep 2020

PONE-D-19-33903R2

Developing political-ecological theory: The need for Many-Task Computing

PLOS ONE

Dear Dr. Haas,

Thank you for submitting your manuscript to PLOS ONE. After careful consideration, we feel that it has merit but does not fully meet PLOS ONE’s publication criteria as it currently stands. Therefore, we invite you to submit a revised version of the manuscript that addresses the points raised during the review process.

The reviewers have commented on their appreciation of the efforts you have made to respond to their comments and the improvement this has made to the manuscript. There remain, however, some important reservations regarding the revised manuscript, and both reviewers have recommended some further changes. Please carefully consider their comment and suggestions, and try to respond not only to the details of their comments but also the intent behind them. An adequate response to the following points will be required for acceptance.

The restructuring to separate the review of related work into a separate section is recommended. This will allow a clearer exposition of the objectives of the work in the introductory section.

Both reviewers have recommended revision of the presentation of the algorithm in Section 2.5.2 to make it more concise and easy to follow, with any extensive commentary removed to supporting text.

The issue remains regarding the consideration given to alternative technologies. Please present the reasons for choosing JavaSpaces clearly and concisely, with relevant comparison with other methods. This has not been achieved with the brief comments already included.

Carefully consider what basic information is really required for a reader interested in this topic.

While you have made commendable efforts to make the manuscript more focussed and concise, the response to the reviewers on the length of the manuscript is unpersuasive. There may be papers of a similar length already published in PLOS ONE, but there are equally many that are shorter, concise and clear. It is these last two characteristics that can significantly enhance a paper.

We look forward to receiving your revised manuscript.

Kind regards,

Andrew Lewis

Academic Editor

PLOS ONE

Reviewers' comments:

Reviewer's Responses to Questions

**Comments to the Author**

1. If the authors have adequately addressed your comments raised in a previous round of review and you feel that this manuscript is now acceptable for publication, you may indicate that here to bypass the “Comments to the Author” section, enter your conflict of interest statement in the “Confidential to Editor” section, and submit your "Accept" recommendation.

Reviewer #3: (No Response)

Reviewer #4: (No Response)

2. Is the manuscript technically sound, and do the data support the conclusions?

Reviewer #3: Yes

Reviewer #4: Partly

3. Has the statistical analysis been performed appropriately and rigorously? 

Reviewer #3: Yes

Reviewer #4: Yes

4. Have the authors made all data underlying the findings in their manuscript fully available?

Reviewer #3: No

Reviewer #4: Yes

5. Is the manuscript presented in an intelligible fashion and written in standard English?

Reviewer #3: Yes

Reviewer #4: Yes

6. Review Comments to the Author

Reviewer #3: The manuscript is certainly a large improvement on what was originally submitted. Just a few things to address:

* My concern is the emphasis on a particular technology, JavaSpaces. It implies that the contribution of the paper is dependant on this technology. There is no harm saying that JavaSpaces was used as the implementation of the ideas. The discussion of the computational model needs to be cast in more general terms. As such, the current Section 2.7 needs to be recast and rewritten. Also, the very basic discussions of runtime systems and compiling etc does not need to be included.

* Section 1 is far too big. The author needs to put the literature review components into a new section (Section 2) and reorganise the remaining components into Section 1.

* Remove paragraphs from the Abstract. It is also too long and needs to be compacted.

* Section 2.5.2 is not an algorithm. An algorithm is a far more precise set of simple steps that is readily translatable into code. The author needs to a) have a textual description under 2.5.2 explaining what is trying to be achieved, and then say the set of high level steps follow. I would encourage the author to try to cast this is an algorithm (there are plenty of papers out there with good examples).

Reviewer #4: Thank you for the changes to the work. Thank you also for using latexdiff to mark up the changes. It has been the task of re-reviewing the work much easier.

Regarding the length, the criticism from me and other reviewers has been that the work was overly long _given the message intended to be delivered_, and that the space devoted to background material hindered delivery of the message. The cuts to Sections 1 and 2 are welcome, and do help focus the paper better, although I strongly suspect it could be further targeted: 13000 words is a _long_ paper, and with no description of how you 'selected' those 5 other PLOS ONE papers, nor of why you consider them to be 'typical' (common, near average, near median?) length for the journal, it's unclear how this justifies the current length.

The core issue from previous revisions remains: is this paper expounding improved statistical modelling and evaluation techniques for this domain, or proposing a computing approach to make them tractable? Trying to do both makes the work very long, and lessens the impact of both parts.

Picking up from another review's comment and the response to it, the algorithm in 2.5.2 is still overly long and, despite the claims to have been made shorter, is actually only one sentence shorter, with several steps consisting of multiple paragraphs. That is not the typical way an algorithm is presented. Steps should be concise and clear, not laden with explanatory text. Consider splitting into an algorithm where, ideally, each step fits on 1-2 lines, with an explanation of the major tasks placed in normal paragraphs nearby.

While it's good that some text has been added to discuss alternatives to JavaSpaces, and the response claims that these alternatives are presented in a way that shows they could be used to implement the approach defined in Section 2, that is unfortunately not how the section reads. Its placement, in the middle of the discussion of JavaSpaces, is odd, since the decision of which technology to use has clearly already been made. It offers little comparison between the chosen technology (remember, we're _not_ asking you to change that) and the alternatives, and is dismissive of Docker, incorrectly tying it to MPI (it actually takes extra effort to make Docker work with MPI; it is not its most natural use). I suggest either: moving the discussion of alternatives, including JavaSpaces, to the start of the section; or adding additional critique and comparison that either (1) indicates which of these alternatives could be used with neither benefit nor loss (i.e., JavaSpaces is one, equally valid implementation choice) or (2) demonstrates why JavaSpaces is a superior choice.

Relatedly, in 3.1, could an estimate of the wallclock time if parallel computing had not been used be given, as well as an indication of the relative scale of the case study compared to other envisaged uses? This would go a long way to justifying 'the need for many-task computing', because as presented it looks like it would take ~45 hours to evaluate a model that would inform planning on a 5-10 year horizon, which is actually not that long to wait for an answer.

In 3.2, what do the parameters in Table 3 represent? It's unclear how the reader is meant to derive much meaning from numbers 168, 165, 171, 174, 183.

The 'additional computational evidence [at] lines 805-811' consists entirely of references to the author's own prior work. Could any other _new_ computational evidence beyond the single case study in this paper be included?

The moved section 4.1 seems remarkably parochial. Australian and New Zealand (presumably European, too) governments have extensive HPC facilities that are free for researcher use and hence, governmental use. Even amongst the commercial HPC providers, products like Microsoft Azure and Amazon EC2 have been completely ignored, yet they have data centres in many locations around the world. I don't object to the inclusion of the section, but if it's to be included at all it should have some more general advice.

7. PLOS authors have the option to publish the peer review history of their article (what does this mean?). If published, this will include your full peer review and any attached files.

Reviewer #3: No

Reviewer #4: No

---

## [Author Response · Author response to Decision Letter 2]

27 Sep 2020

Please see my Response to Reviewers.

---

## [Decision Letter · Decision Letter 3]

30 Oct 2020

Developing political-ecological theory: The need for Many-Task Computing

PONE-D-19-33903R3

Dear Dr. Haas,

We’re pleased to inform you that your manuscript has been judged scientifically suitable for publication and will be formally accepted for publication once it meets all outstanding technical requirements.

Kind regards,

Andrew Lewis

Academic Editor

PLOS ONE

Additional Editor Comments (optional):

Reviewers' comments:

Reviewer's Responses to Questions

**Comments to the Author**

1. If the authors have adequately addressed your comments raised in a previous round of review and you feel that this manuscript is now acceptable for publication, you may indicate that here to bypass the “Comments to the Author” section, enter your conflict of interest statement in the “Confidential to Editor” section, and submit your "Accept" recommendation.

Reviewer #3: (No Response)

Reviewer #4: All comments have been addressed

2. Is the manuscript technically sound, and do the data support the conclusions?

Reviewer #3: Yes

Reviewer #4: Yes

3. Has the statistical analysis been performed appropriately and rigorously? 

Reviewer #3: Yes

Reviewer #4: Yes

4. Have the authors made all data underlying the findings in their manuscript fully available?

Reviewer #3: Yes

Reviewer #4: Yes

5. Is the manuscript presented in an intelligible fashion and written in standard English?

Reviewer #3: Yes

Reviewer #4: Yes

6. Review Comments to the Author

Reviewer #3: I have now been through a few iterations of this paper. The author has been addressing my concerns each time. I believe it is at a point where it is acceptable.

Reviewer #4: It's clear a lot of careful work has gone in to preparing this revision, and it addresses the majority of my concerns about the paper. The use of JavaSpaces is now much more thoroughly justified and the background work easier to follow. It's time for the work to be available for examination by a wider audience.

7. PLOS authors have the option to publish the peer review history of their article (what does this mean?). If published, this will include your full peer review and any attached files.

Reviewer #3: No

Reviewer #4: No

---

## [Editor Report · Acceptance letter]

12 Nov 2020

PONE-D-19-33903R3 

Developing political-ecological theory: The need for Many-Task Computing 

Dear Dr. Haas:

I'm pleased to inform you that your manuscript has been deemed suitable for publication in PLOS ONE. Congratulations! Your manuscript is now with our production department. 

Kind regards, 

on behalf of

Dr. Andrew Lewis 

Academic Editor

PLOS ONE